# Innovative Line for Door Production TechnoPORTA—Technological and Economic Aspects of Application of Wood-Based Materials

Zdzisław Kwidziński [1], Joanna Bednarz [2], Marta Pędzik [3,4], Łukasz Sankiewicz [1], Piotr Szarowski [1], Bartłomiej Knitowski [1] and Tomasz Rogoziński [4,*]

[1] Porta KMI Poland, Szkolna 54, 84-239 Bolszewo, Poland; Zdzislaw_Kwidzinski@porta.com.pl (Z.K.); Lukasz_Sankiewicz@porta.com.pl (Ł.S.); Piotr_Szarowski@porta.com.pl (P.S.); Bartlomiej_Knitowski@porta.com.pl (B.K.)

[2] Department of International Business, Faculty of Economics, Univesity of Gdańsk, Armii Krajowej 119/121, 81-824 Sopot, Poland; Joanna.bednarz@ug.edu.pl

[3] Łukasiewicz Research Network—Wood Technology Institute, Winiarska 1, 60-654 Poznań, Poland; marta.pedzik@itd.lukasiewicz.gov.pl

[4] Department of Furniture Design, Faculty of Forestry and Wood Technology, Poznań University of Life Sciences, Wojska Polskiego 38/42, 60-627 Poznań, Poland

* Correspondence: tomasz.rogozinski@up.poznan.pl

**Abstract:** Material losses are caused by the machining process and the manufacturing process, as well as the use of excessive dimensional allowances applied to the machined materials. An adequate reduction of the planned machining allowances for wood-based panel components is possible when the machining line is properly designed and equipped with high-precision machines and devices. The aim of the study was to determine the size of material savings in relation to the most important construction materials when implementing an innovative technological line for processing industrial doors made of wood materials. The achieved savings improve the competitiveness of the door manufacturer on the market. In order to calculate the material savings obtained in the production of the most important models of door leaves that can be obtained in machining on the TechnoPORTA line, numerical data were compiled specifying the dimensions of semi-finished products, taking into account machining allowances applied before and after reduction. The implementation of the TechnoPORTA line makes it possible to reduce the negative impact on the environment by reducing the consumption of wood. It reduces the consumption of materials and the operating costs associated with the reduction of labor intensity, the load on machines and devices, and inventory levels.

**Keywords:** allowance; competitive advantage; mass customization; machining

## 1. Introduction

In enterprises dealing with processing and machining of wood or wood-based materials, including boards and glued laminated timber, a significant part of variable costs are the costs of purchase of these materials. As a result of technological operations, a large amount of post-production waste is generated, so it is important to reduce it [1]. The difference between the initial volume of raw material and the volume of finished items is often material waste. Material losses are caused by the machining process and the manufacturing process, as well as the use of excessive dimensional allowances applied to the machined materials, which is forced by the frequently changing dimensions of wood due to, among other things, drying [2]. The easiest way to determine the value of the machining allowance is to relate the dimensions of the blank to the previous operation. Excess material allowances generate material losses and higher manufacturing costs, including for energy consumption [3]. Losses of materials result also from high variability of manufactured products, which

concerns mainly the enterprises with customized production; therefore, it is particularly important in such a case to control the parameters of production and product quality.

During drying, lumber shrinks under the influence of moisture changes, but dimensional changes do not occur uniformly in all anatomical directions. The accuracy of cutting, even with high-quality saws, is largely dependent on individual defects in the wood and its anisotropic structure [4–6]. This problem is to some extent eliminated with the use of wood-based panels. When sawing the boards along a curve, the stiffness of the saw blade makes it difficult to maintain a constant width of the cut, resulting in changes in longitudinal dimensions and different thicknesses of the resulting pieces [7]. In addition, operations such as planning and grinding also require material allowances. Therefore, for this and other operations that interfere with the target dimension of the material, it is important to leave some margin of safety [1]. This results in loss of material, which translates into costs for companies and problems with large amounts of waste that have to be disposed of. Dust waste generated during mechanical processing is also problematic, due to the high harmfulness to the health of slinger workers, and threatens the proper operation of machines and devices [8–11]. Determining optimal machining conditions in the design of a process operation poses significant technological and organizational challenges [12–14]. Therefore, machining accuracy is an indispensable factor in achieving high product quality while achieving the highest possible material efficiency.

An adequate reduction of the planned machining allowances for wood-based panel components is possible when the machining line is properly designed and equipped with high-precision machines and devices. A lack of proper stability, vibrations, tool jumps, or tool wear can lead to unevenness and deviation from the desired dimensions of the finished product. This results in both material and economic losses, as well as production downtime and the need to correct errors. One possible way to reduce the losses incurred may be an investment in the form of automation of production lines. Automating production allows for increased production efficiency while reducing production costs and not incurring additional material costs [15]. Accurate determination of machining allowances is a complex process due to the varying properties of the materials being machined, but is crucial in any technological process [16–18]. The basis for assessing the effectiveness of any change in the technological process is the analysis of technical and economic indicators based on the costs incurred by the production plant.

The technology of production of industrial leaves in Porta KMI Poland S.A. is currently taking place with the use of implemented innovative technological line TechnoPORTA. Technically, the TechnoPORTA line is organized as shown in Figure 1.

The TechnoPORTA line includes the following sections:

- A section for formatting, milling, and edge banding (Module 01). These are the machining stations where the final dimensions of the door leaves are given. Due to the machining accuracy achieved by the machines used, the size of machining allowances could be reduced.
- A postforming section (Module 02).
- A CNC section, drilling, and milling of hinges and locks (Module 03).
- A CNC section, individual milling (Module 04).
- A fitting section (Module 05).
- The line performs the following processes:
- automatic production of industrial door leaves, including products with increased overall dimensions and weight;
- automatic processing technology for narrow edges of industrial door leaves;
- automatic process of technological and production handling of atypical orders.

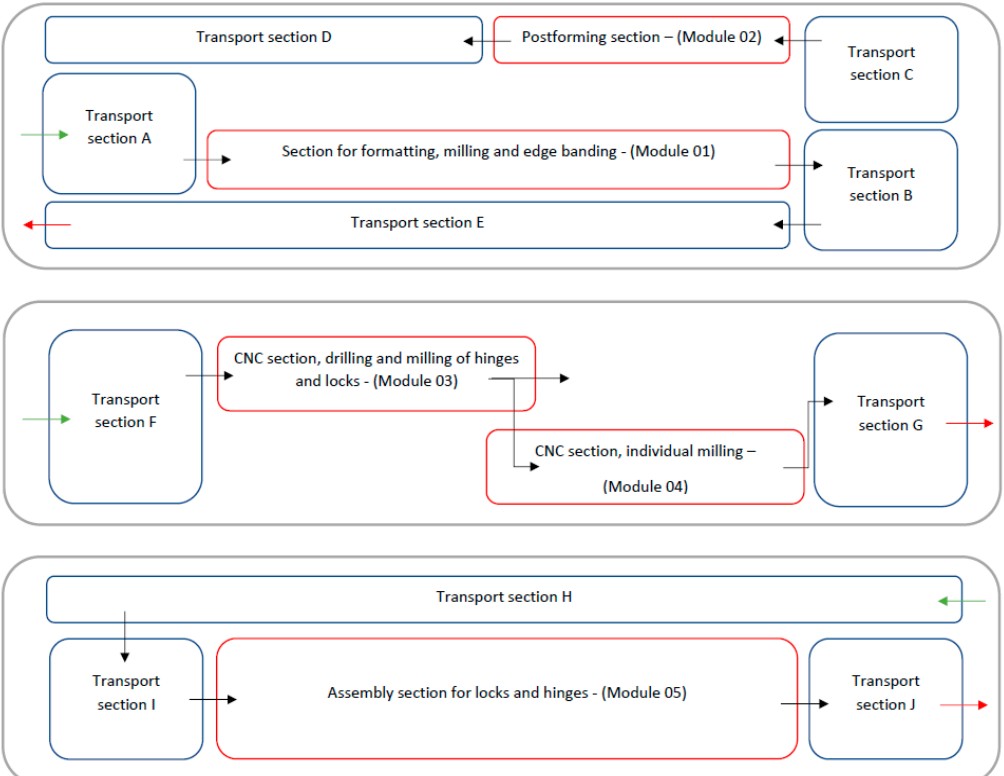

**Figure 1.** Schematic setup of the TechnoPORTA technological line.

The technological line to ensure the best quality and efficiency of production in the conditions of mass customization, where the size of the production series (one door) must meet the highest technical requirements. TechnoPORTA was built with the use of machines and devices characterized by a very high level of general technology. In addition, to ensure the required quality of processing, automatic material positioning, and feeding and parametric control of the operation and changeovers of the machines were used at each stage. This is especially vital for Module 1, related to formatting, milling, and edge banding, as this is where the final dimensions of the door are obtained (Figure 2).

The developed technology is to enable maximum expansion of the door leaf personalization possibilities in accordance with the principle that a production series is one piece of a product. The production of doors is carried out in a maximally automated way, which guarantees a repeatable high quality of products. The design capacity of the developed production line is about 800 pieces of industrial door leaves per one production shift. For the production of doors, wood and wood-composite materials are used as construction materials [19,20]. This is primarily coniferous wood—pine. In addition, smaller amounts of hardwood—oak—are also used. The most important wood composites are particleboard, plywood, and fiberboard for door filling and HDF (high density fiberboard) for door cladding. MDF (medium density fiberboard) is also used in the production of frames of some leaf constructions. Wood materials, which are the most important component of the door construction, are unfortunately in short supply and their price is constantly increasing despite the increase in the wood harvest [21,22]. According to statistical data, the area of forest crops in Poland has been increasing for many years, the forest cover of the country is 29.6%. In 2020 the timber coarse wood resources amounted to 2730 million m$^3$, which is an increase of ca. 15% compared to 2010, while compared to 2000 it is almost 37% [23]. Nevertheless, the demand for wood and all kinds of wood products that are the product of its processing, including sawn timber, joinery products, fiberboard and particleboard, as well as furniture, is also constantly increasing [24]. In 2010, the State Forests in Poland sold about 33,731 thousand m3 of timber in total; in 2019 it was already about 41,076 thousand

$m^3$ [23]. At the same time it should be mentioned that the average price of large-sized general-purpose coniferous timber has increased by about 5 PLN/$m^3$ in 3 years [25].

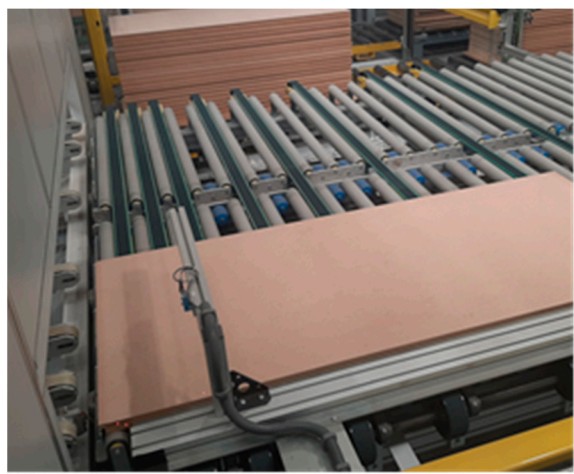
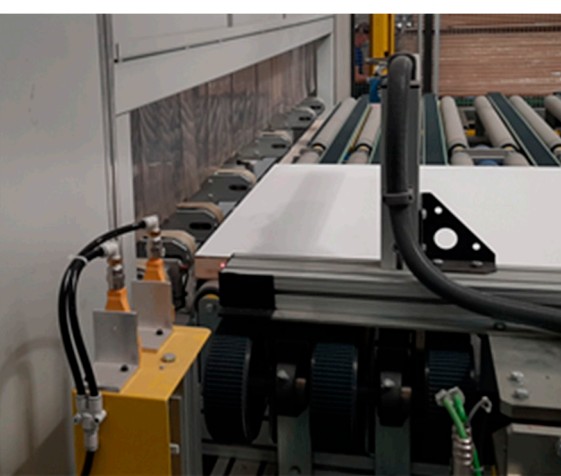

Initial positioning

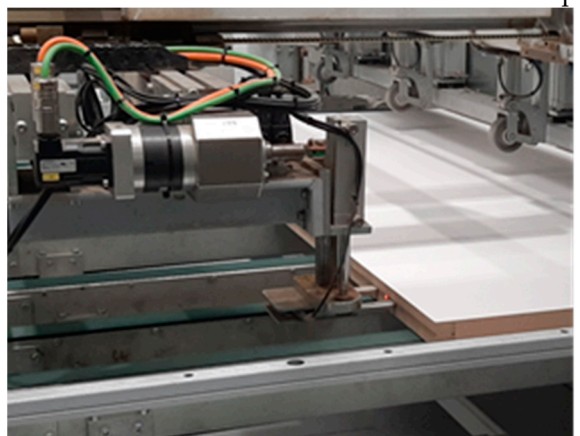

Positioning at short-edge milling

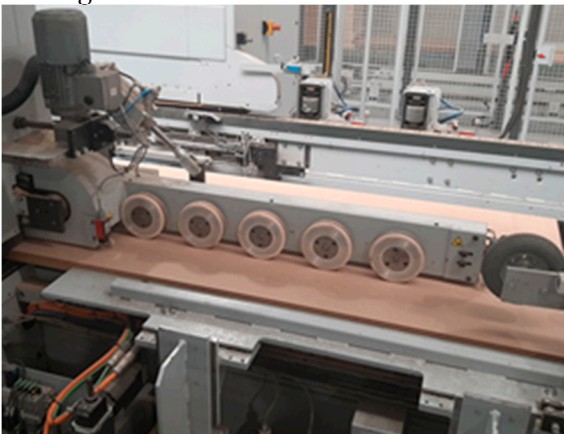

Positioning at long-edge milling

**Figure 2.** Positioning at edge milling.

According to the Oslo Manual, a business innovation "is a new or improved product or business process (or combination thereof) that differs significantly from the firm's previous products or business processes and that has been introduced on the market or brought into use by the firm" [26]. Business processes should be understood as those processes that "include all core activities by the firm to produce products and all ancillary or supporting activities." The TechnoPORTA technological line is consistent with all eight broad types of activities in pursuit of innovation: research and experimental development (R&D) activities, engineering, design and other creative work activities, marketing and brand equity activities, IP-related activities, employee training activities, software development and database activities, activities related to the acquisition or lease of tangible assets, and innovation management activities [26]. Thanks to the implemented innovations it is possible to introduce technological and constructional changes in products, which will allow the company to increase its competitiveness.

## 2. Aim of the Study

The aim of the study was to determine the size of material savings in relation to the most important construction materials when implementing an innovative technological line for processing industrial leaves made of wood materials. Additionally, the paper describes the possibilities of beneficial influence on building the competitive advantage of Porta KMI Poland S.A. by reduction of a primary factor-material cost in door-leaf production.



## 3. Materials and Methods

In order to carry out calculations of achieved material savings in the production of door leaves on the TechnoPORTA line, construction data were obtained in the form of sheets (Tables 1–4, numbering of elements is in accordance with the numbering in Figures 3–6) containing numerical data defining the dimensions of semi-finished products, taking into consideration machining allowances used previously and after reduction, achievable in machining on the TechnoPORTA line. The data included four constructions comprising the most important models of leaves manufactured in accordance with dimensional requirements contained in PN (Polish standard). These constructions are numbered 1, 2, 3, 4 in order.

**Table 1.** Construction 1. Previously used values of machining allowances; reduced machining allowances, variant 1; variant 2.

| No. | Element | Material | Length According to the Width of the Leaf | | | Width × Thickness | Pieces |
|---|---|---|---|---|---|---|---|
| | | | **80** | **90** | **100** | | |
| 1 | vertical stile | soft wood—pine | 2038 | 2038 | 2038 | 96 × 43 | 2 |
| 2 | horizontal stile | soft wood—pine | 686 | 810 | 886 | 96 × 43 | 2 |
| 3 | door filling | solid particleboard | 685 | 809 | 885 | 1870 × 12 | 2 |
| 4 | door filling | fiberboard | 656 | 780 | 856 | 1842 × 12 | 1 |
| 5 | door filling | HDF | 656 | 780 | 856 | 1842 × 3 | 2 |
| 6 | cladding | HDF | 856 | 984 | 1056 | 2046 × 3 | 2 |
| 7 | top layer | laminate CPL | 856 | 984 | 1056 | 2046 × 0.2/0.7 | 2 |
| No. | Element | Material | Length According to the Width of the Leaf | | | Width × Thickness | Pieces |
| | | | 80 | 90 | 100 | | |
| 1 | vertical stile | soft wood—pine | 2038 | 2038 | 2038 | 94 × 43 | 2 |
| 2 | horizontal stile | soft wood—pine | 690 | 814 | 890 | 94 × 43 | 2 |
| 3 | door filling | solid particleboard | 689 | 813 | 889 | 1874 × 12 | 2 |
| 4 | door filling | fiberboard | 660 | 784 | 860 | 1846 × 12 | 1 |
| 5 | door filling | HDF | 660 | 784 | 860 | 1846 × 3 | 2 |
| 6 | cladding | HDF | 856 | 984 | 1056 | 2046 × 3 | 2 |
| 7 | top layer | laminate CPL | 856 | 984 | 1056 | 2046 × 0.2/0.7 | 2 |
| No. | Element | Material | Length According to the Width of the Leaf | | | Width × Thickness | Pieces |
| | | | 80 | 90 | 100 | | |
| 1 | vertical stile | soft wood—pine | 2038 | 2038 | 2038 | 96 × 43 | 2 |
| 2 | horizontal stile | soft wood—pine | 686 | 810 | 886 | 96 × 43 | 2 |
| 3 | door filling | solid particleboard | 685 | 809 | 885 | 1870 × 12 | 2 |
| 4 | door filling | fiberboard | 656 | 780 | 856 | 1842 × 12 | 1 |
| 5 | door filling | HDF | 656 | 780 | 856 | 1842 × 3 | 2 |
| 6 | cladding | HDF | 856 | 984 | 1056 | 2046 × 3 | 2 |
| 7 | top layer | laminate CPL | 856 | 984 | 1056 | 2046 × 0.2/0.7 | 2 |

From the difference in dimensions of semi-finished products having previously used values of machining allowances and semi-finished products with reduced allowances used in the TechnoPORTA line, material savings were calculated in two variants. In the first variant, it was assumed that the reduced machining allowance would concern the dimension of the frame width. In the second variant, it was assumed that the reduction in the dimensions of the semi-finished product would result from the reduced dimensions of the input. As a result, less material will be used for the manufacture of the frames or the input, respectively.

**Table 2.** Construction 2. Previously used values of machining allowances; reduced machining allowances, variant 1; variant 2.

| No. | Element | Material | Length According to the Width of the Leaf | | | | | Width × Thickness | Pieces |
|---|---|---|---|---|---|---|---|---|---|
| | | | 60 | 70 | 80 | 90 | 100 | | |
| 1 | vertical stile | soft wood—pine | 2038 | 2038 | 2038 | 2038 | 2038 | 44 × 33.1 | 2 |
| 2 | top horizontal stile | soft wood—pine | 564 | 664 | 764 | 864 | 964 | 44 × 33.1 | 1 |
| 3 | bottom horizontal stile | soft wood—pine | 564 | 664 | 764 | 864 | 964 | 92 × 33.1 | 1 |
| 4 | door filling | solid particle-board/tubular particleboard | 400 | 500 | 600 | 700 | 800 | 1897 × 33 | 1 |
| 5 | bracing vertical stile | solid particle-board/tubular particleboard | 1897 | 1897 | 1897 | 1897 | 1897 | 60 × 33 | 2 |
| 6 | bracing vertical stile | plywood | 1897 | 1897 | 1897 | 1897 | 1897 | 18 × 33 | 2 |
| 7 | cladding | HDF | 660 | 760 | 860 | 960 | 1060 | 2050 × 3 | 2 |
| 8 | top layer | laminate CPL, HPL, finish foil | 660 | 760 | 860 | 960 | 1060 | 2050 × 0.15/0.2/0.7 | 2 |
| No. | Element | Material | Length According to the Width of the Leaf | | | | | Width × Thickness | Pieces |
| | | | 60 | 70 | 80 | 90 | 100 | | |
| 1 | vertical stile | soft wood—pine | 2036 | 2036 | 2036 | 2036 | 2036 | 43 × 33.1 | 2 |
| 2 | top horizontal stile | soft wood—pine | 564 | 664 | 764 | 864 | 964 | 43 × 33.1 | 1 |
| 3 | bottom horizontal stile | soft wood—pine | 564 | 664 | 764 | 864 | 964 | 91 × 33.1 | 1 |
| 4 | door filling | solid particle-board/tubular particleboard | 400 | 500 | 600 | 700 | 800 | 1897 × 33 | 1 |
| 5 | bracing vertical stile | solid particle-board/tubular particleboard | 1897 | 1897 | 1897 | 1897 | 1897 | 60 × 33 | 2 |
| 6 | bracing vertical stile | plywood | 1897 | 1897 | 1897 | 1897 | 1897 | 18 × 33 | 2 |
| 7 | cladding | HDF | 658 | 758 | 858 | 958 | 1058 | 2048 × 3 | 2 |
| 8 | top layer | laminate CPL, HPL, finish foil | 658 | 758 | 858 | 958 | 1058 | 2048 × 0.15/0.2/0.7 | 2 |
| No. | Element | Material | Length According to the Width of the Leaf | | | | | Width × Thickness | Pieces |
| | | | 60 | 70 | 80 | 90 | 100 | | |
| 1 | vertical stile | soft wood—pine | 2036 | 2036 | 2036 | 2036 | 2036 | 44 × 33.1 | 2 |
| 2 | top horizontal stile | soft wood—pine | 562 | 662 | 762 | 862 | 962 | 44 × 33.1 | 1 |
| 3 | bottom horizontal stile | soft wood—pine | 562 | 662 | 762 | 862 | 962 | 92 × 33.1 | 1 |
| 4 | door filling | solid particle-board/tubular particleboard | 398 | 498 | 598 | 698 | 798 | 1895 × 33 | 1 |
| 5 | bracing vertical stile | solid particle-board/tubular particleboard | 1895 | 1895 | 1895 | 1895 | 1895 | 60 × 33 | 2 |
| 6 | bracing vertical stile | plywood | 1895 | 1895 | 1895 | 1895 | 1895 | 18 × 33 | 2 |
| 7 | cladding | HDF | 658 | 758 | 858 | 958 | 1058 | 2048 × 3 | 2 |
| 8 | top layer | laminate CPL, HPL, finish foil | 658 | 758 | 858 | 958 | 1058 | 2048 × 0.15/0.2/0.7 | 2 |

**Table 3.** Construction 3. Previously used values of machining allowances; reduced machining allowances, variant 1.

| No. | Element | Material | Length According to the Width of the Leaf | | | | | Width × Thickness | Pieces |
|---|---|---|---|---|---|---|---|---|---|
| | | | 60 | 70 | 80 | 90 | 100 | | |
| 1 | vertical stile | MDF | 2038 | 2038 | 2038 | 2038 | 2038 | 35 × 33.1 | 2 |
| 2 | top horizontal stile | MDF | 580 | 680 | 780 | 880 | 980 | 35 × 33.1 | 1 |
| 3 | bottom horizontal stile | soft wood—pine | 580 | 680 | 780 | 880 | 980 | 35 × 33.1 | 1 |
| 4 | interior vertical stile | solid particleboard | 1664 | 1664 | 1664 | 1664 | 1664 | 22 × 33.1 | 1 |
| 5 | filling for lock | soft wood—pine/MDF | 320 | 320 | 320 | 320 | 320 | 70 × 33.1 | 2 |
| 6 | door filling | honeycomb | - | - | - | - | - | - | 0 |
| 7 | cladding | HDF | 660 | 760 | 860 | 960 | 1060 | 2050 × 3 | 2 |
| No. | Element | Material | Length According to the Width of the Leaf | | | | | Width × Thickness | Pieces |
| | | | 60 | 70 | 80 | 90 | 100 | | |
| 1 | vertical stile | MDF | 2036 | 2036 | 2036 | 2036 | 2036 | 34 × 33.1 | 2 |
| 2 | top horizontal stile | MDF | 580 | 680 | 780 | 880 | 980 | 34 × 33.1 | 1 |
| 3 | bottom horizontal stile | soft wood—pine | 580 | 680 | 780 | 880 | 980 | 34 × 33.1 | 1 |
| 4 | interior vertical stile | solid particleboard | 1664 | 1664 | 1664 | 1664 | 1664 | 22 × 33.1 | 1 |
| 5 | filling for lock | soft wood—pine/MDF | 320 | 320 | 320 | 320 | 320 | 70 × 33.1 | 2 |
| 6 | door filling | honeycomb | - | - | - | - | - | - | 0 |
| 7 | cladding | HDF | 658 | 758 | 858 | 958 | 1058 | 2048 × 3 | 2 |

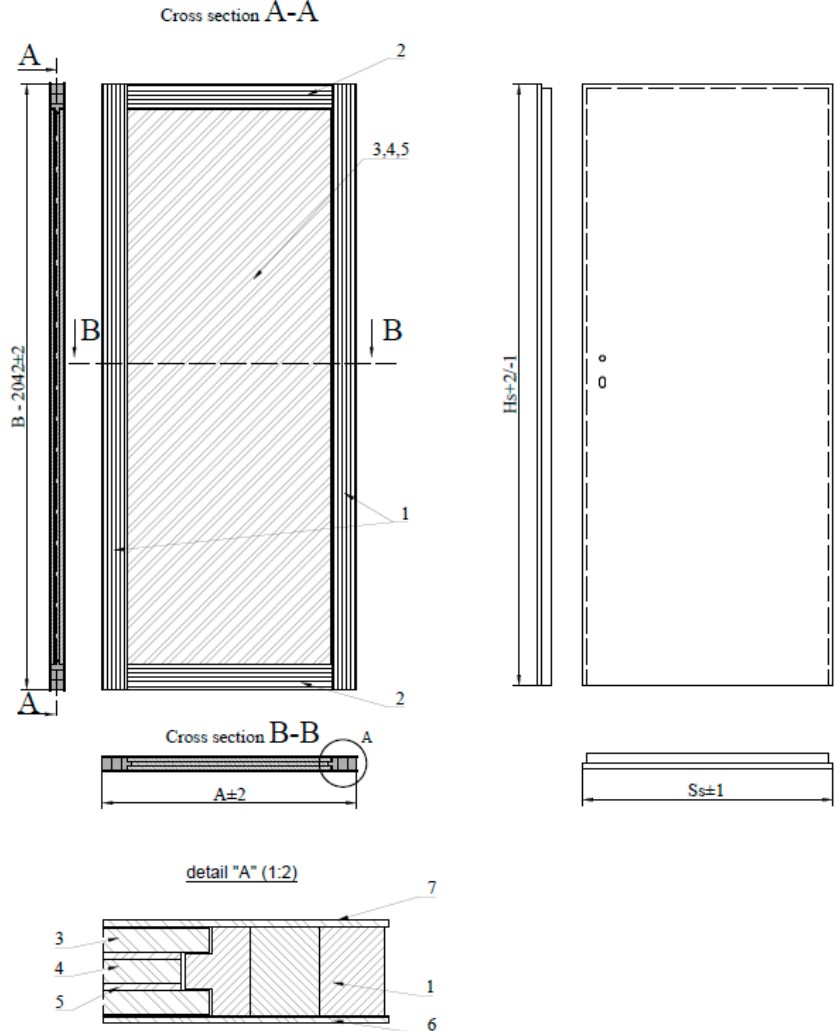

**Figure 3.** Door construction 1.

**Table 4.** Construction 4. Previously used values of machining allowances; reduced machining allowances, variant 1; variant 2.

| No. | Element | Material | Length According to the Width of the Leaf | | | | | Width × Thickness | Pieces |
|---|---|---|---|---|---|---|---|---|---|
| | | | 60 | 70 | 80 | 90 | 100 | | |
| 1 | vertical stile | hard wood—oak | 2042 | 2042 | 2042 | 2042 | 2042 | 46 × 37.1 | 2 |
| 2 | top horizontal stile | hard wood—oak | 562 | 662 | 762 | 870 | 962 | 46 × 37.1 | 1 |
| 3 | bottom horizontal stile internal | soft wood—pine | 562 | 662 | 762 | 870 | 962 | 46 × 37.1 | 1 |
| 4 | bottom horizontal stile external | hard wood—oak | 562 | 662 | 762 | 870 | 962 | 46 × 37.1 | 1 |
| 5 | door filling | solid particleboard | 560 | 660 | 760 | 868 | 960 | 1902 × 12 | 3 |
| 6 | cladding | HDF | 660 | 760 | 860 | 966 | 1060 | 2050 × 3 − 5 | 2 |
| 7 | top layer | laminate CPL | 660 | 760 | 860 | 966 | 1060 | 2050 × 0.2 − 1 | 2 |
| No. | Element | Material | Length According to the Width of the Leaf | | | | | Width × Thickness | Pieces |
| | | | 60 | 70 | 80 | 90 | 100 | | |
| 1 | vertical stile | hard wood—oak | 2038 | 2038 | 2038 | 2038 | 2038 | 44 × 37.1 | 2 |
| 2 | top horizontal stile | hard wood—oak | 562 | 662 | 762 | 870 | 962 | 44 × 37.1 | 1 |
| 3 | bottom horizontal stile internal | soft wood—pine | 562 | 662 | 762 | 870 | 962 | 45 × 37.1 | 1 |
| 4 | bottom horizontal stile external | hard wood—oak | 562 | 662 | 762 | 870 | 962 | 45 × 37.1 | 1 |
| 5 | door filling | solid particleboard | 560 | 660 | 760 | 868 | 960 | 1902 × 12 | 3 |
| 6 | cladding | HDF | 656 | 756 | 856 | 962 | 1056 | 2046 × 3 − 5 | 2 |
| 7 | top layer | laminate CPL | 656 | 756 | 856 | 962 | 1056 | 2046 × 0.2 − 1 | 2 |
| No. | Element | Material | Length According to the Width of the Leaf | | | | | Width × Thickness | Pieces |
| | | | 60 | 70 | 80 | 90 | 100 | | |
| 1 | vertical stile | hard wood—oak | 2038 | 2038 | 2038 | 2038 | 2038 | 46 × 37.1 | 2 |
| 2 | top horizontal stile | hard wood—oak | 558 | 658 | 758 | 866 | 958 | 46 × 37.1 | 1 |
| 3 | bottom horizontal stile internal | soft wood—pine | 558 | 658 | 758 | 866 | 958 | 46 × 37.1 | 1 |
| 4 | bottom horizontal stile external | hard wood—oak | 558 | 658 | 758 | 866 | 958 | 46 × 37.1 | 1 |
| 5 | door filling | solid particleboard | 556 | 656 | 756 | 864 | 956 | 1898 × 12 | 3 |
| 6 | cladding | HDF | 656 | 756 | 856 | 962 | 1056 | 2046 × 3 − 5 | 2 |
| 7 | top layer | laminate CPL | 656 | 756 | 856 | 962 | 1056 | 2046 × 0.2 − 1 | 2 |

Construction 1 (Figure 2) contained an industrial leaf filling consisting of a special multi-layer structure with wood composites in a softwood plywood frame. The entire structure was clad with an HDF. Thanks to the use of aluminum sheet, exceptional resistance to unfavorable conditions of use, characteristic of a staircase; high temperature difference; and high air humidity were obtained.

In construction 2 (Figure 3), the most significant feature was the additional reinforcement with an internal frame. The leaf cladding consisted of a layer of aluminum and wood-based HDF. The main frame of the leaf was made of softwood plywood. These are anti-burglary doors, which have a sound insulation class of $R_w$ = 32 dB and resistance to burglary—provided that the leaf is completed with a dedicated steel frame and descending threshold.

The solution used in construction 3 (Figure 4) was an eco-friendly honeycomb filling—reusable, recyclable. The whole was covered with HDF coated with an acrylic UV paint with very high resistance to dirt (coffee, alcohol, grease), abrasion, scratching, and impact.

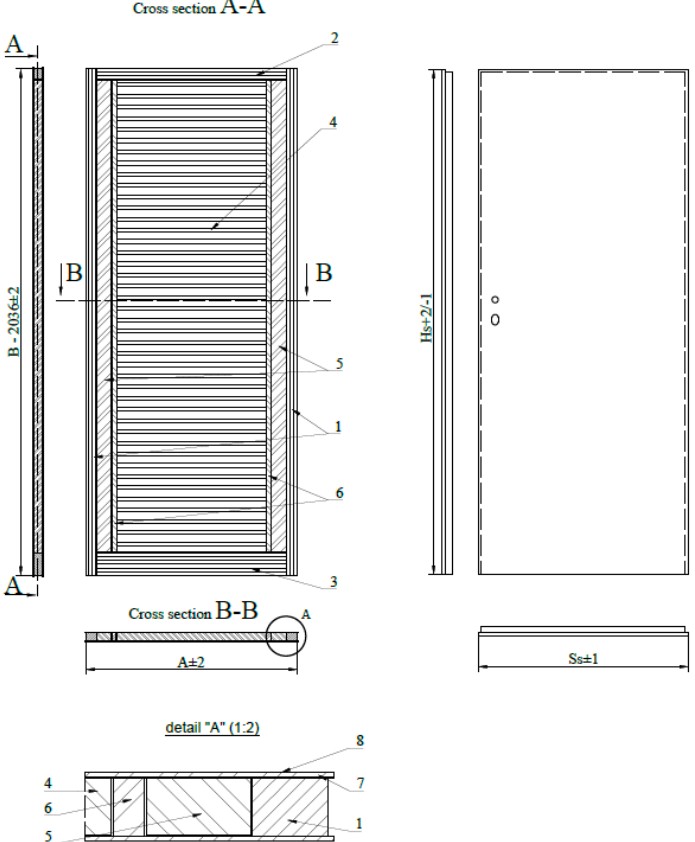

**Figure 4.** Door construction 2.

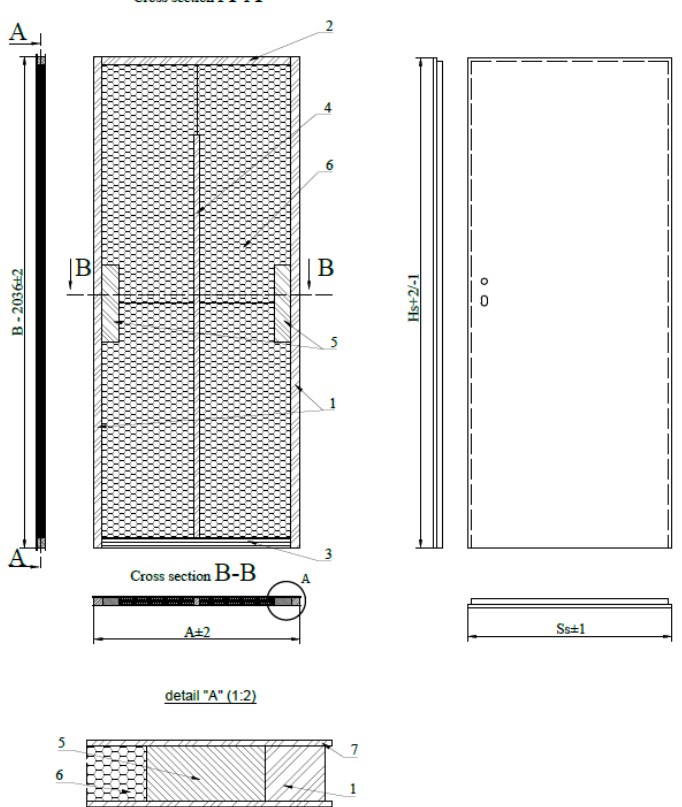

**Figure 5.** Door construction 3.

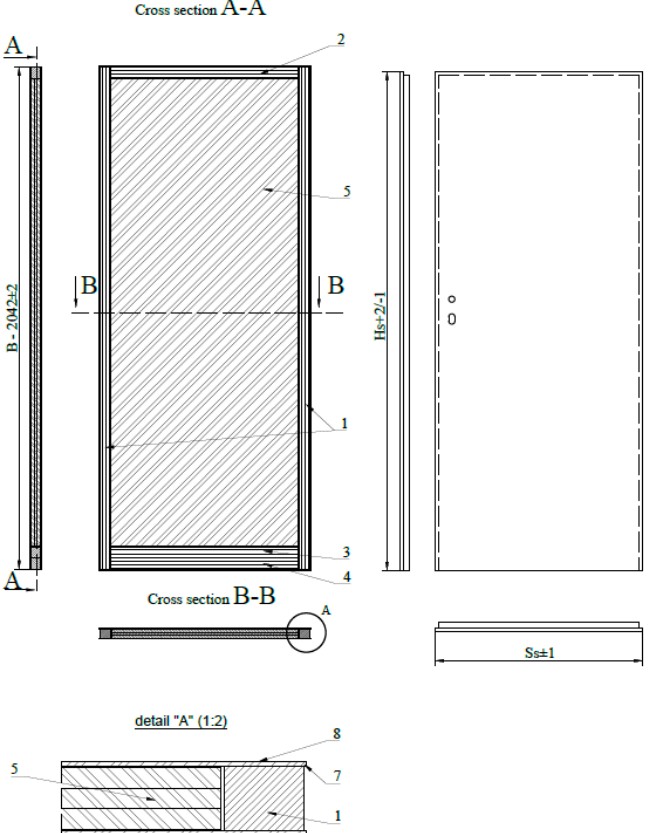

**Figure 6.** Door construction 4.

Construction 4 (Figure 5) is distinguished by the fact that the filling of the leaf was a special five-layer Fireproof PORTA structure in a frame made of softwood plywood. The entire structure was covered with HDF board. The sides of the leaf were covered with ABS (acrylonitrile–butadiene–styrene) tape and were given a seal that swells when exposed to fire.

The values of material savings were calculated per 1000 leaves. With known volumes of actual production in 2020 (Table 5), material savings were calculated on an annual basis for major construction materials, taking into account the quantitative proportions of the production volume of individual door constructions over the entire quantity of products manufactured in 2020.

**Table 5.** Production volume of particular door constructions in the TechnoPORTA line in 2020.

| Construction No. | | | |
|---|---|---|---|
| 1 | 2 | 3 | 4 |
| Volume of pieces/Year | | | |
| 24,000 | 134,000 | 62,000 | 10,000 |

The final amount of achievable wood savings also takes into account the material efficiency of the pretreatment. Because of the two-stage processing—obtaining blanks (battens from lumber) and milling the leaves—the material efficiency index of the pretreatment must be considered when estimating savings. With respect to wood species used in door manufacturing, a satisfactory value of this index should be about 75% [27,28]. Thus, the global annual wood savings can be calculated as follows:

$$\Delta M = \frac{\Delta S}{RMEI} \left[ m^3/year \right] \tag{1}$$

where:

$\Delta S$—annual wood savings due to reduction of machining allowance, m$^3$/year;
*RMEI*—raw material efficiency index = 0.75.

## 4. Results

The raw material savings for the individual constructions are shown in the graphs (Figures 7–10) taking into account both variants of reduction of machining allowances. This mainly concerns savings in wood and MDF obtained by reducing the width of the frames in variant 1, and savings in wood-based panels obtained by reducing the height and width of the input panels in variant 2. The savings are presented both in absolute terms and as a percentage of the amount of material used in the existing technology.

Variant 1

Pine wood

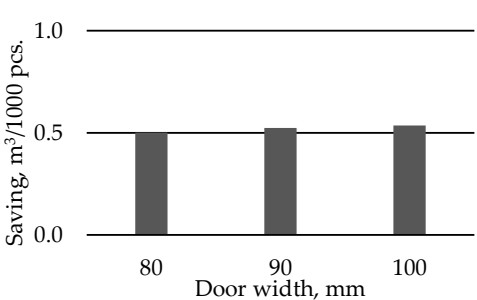
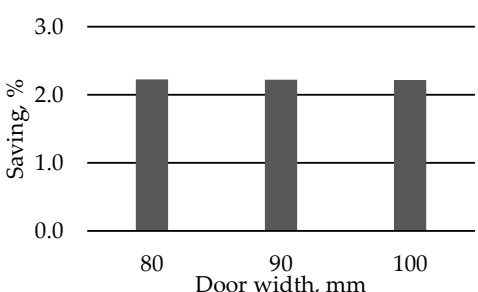

Variant 2

particleboard

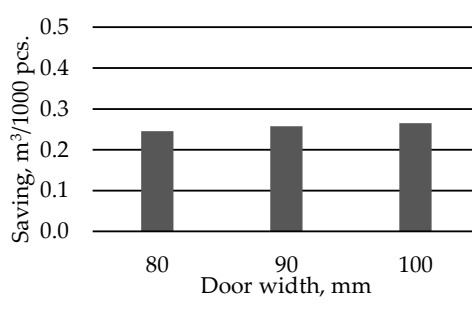
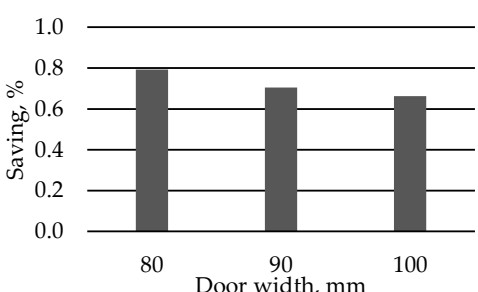

fiberboard

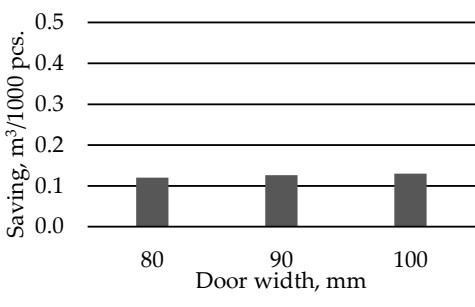
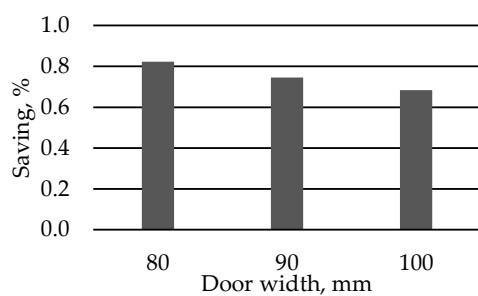

HDF

**Figure 7.** *Cont.*

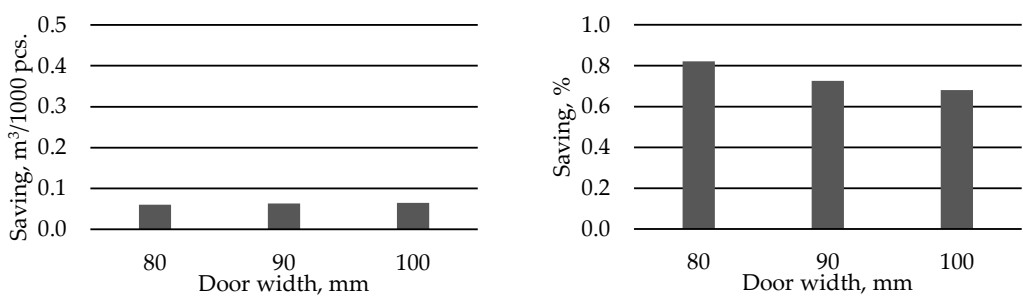

**Figure 7.** Material savings obtained by reducing the machining allowance in construction 1.

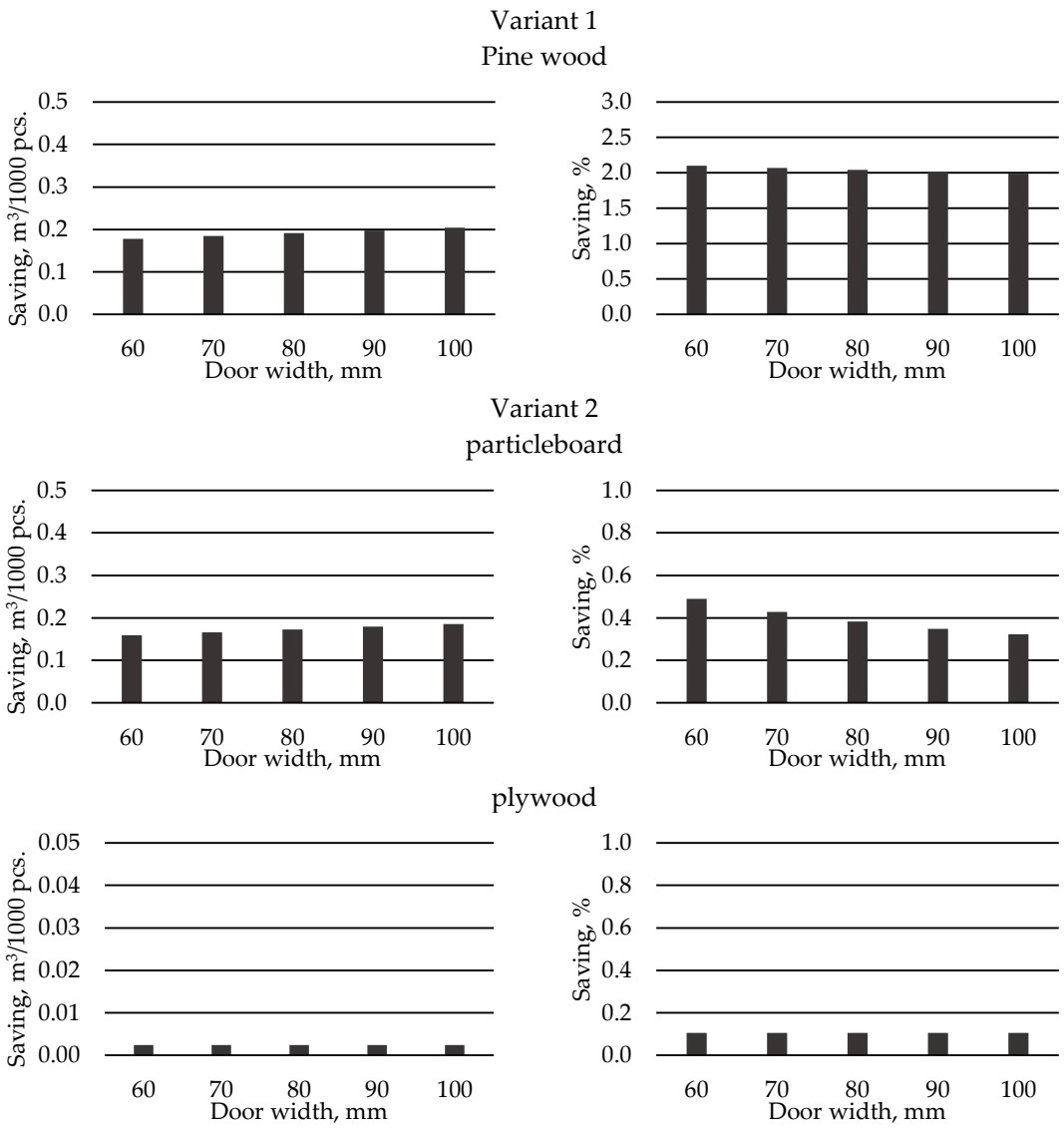

**Figure 8.** Material savings obtained by reducing the machining allowance in construction 2.

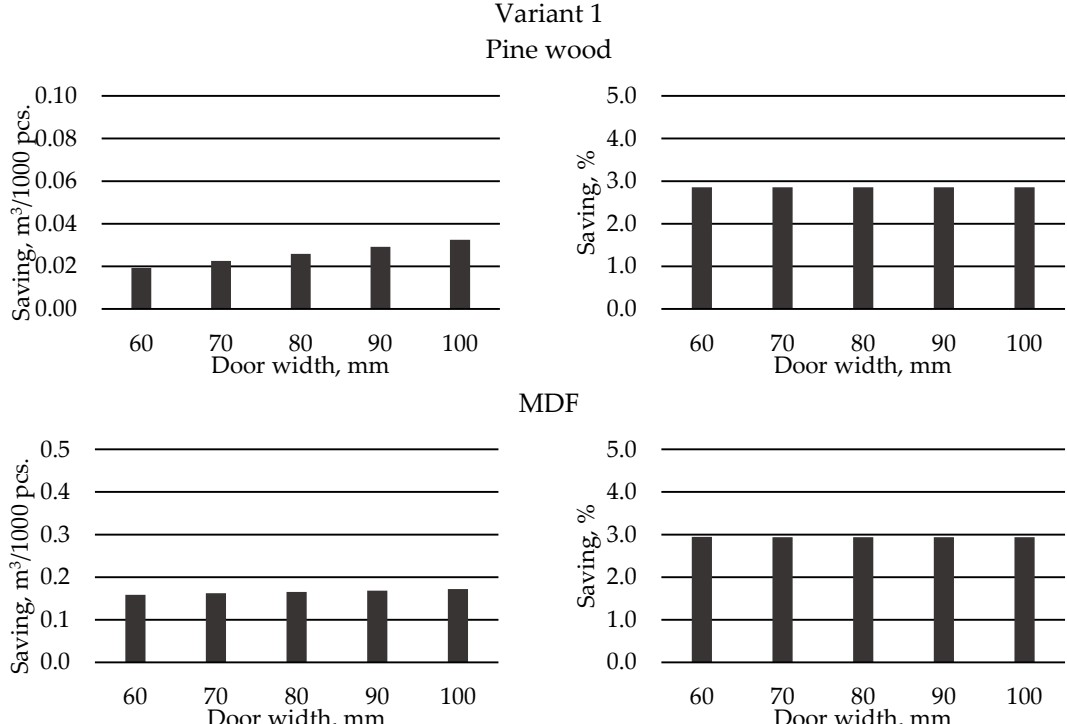

**Figure 9.** Material savings obtained by reducing the machining allowance in construction 3.

Due to the fact that construction 1 is based on wide, wooden frames, it was possible to achieve a large reduction in machining allowance by as much as 4 mm. Therefore, the saving of wood in variant 1 amounted to over 0.5 m³/1000 pieces. The larger the leaf dimensions, the slightly greater the saving. In percentage terms, this results in a 2.2% reduction in frame volume.

In variant 2, when reducing the machining allowance by decreasing the input dimensions, savings of approximately 0.250, 0.125, and 0.06 m³/1000 pieces were obtained for particleboard, plywood, and HDF filling materials, respectively. This saving is also slightly related to the leaf dimensions. Its absolute value increases for increasingly larger leaves, while in percentage terms, as the allowance is not related to leaf dimensions, it decreases from about 0.80% for the smallest leaves to about 0.68% for the largest ones.

In construction 2, invariant 1 of determining the machining allowance, the allowance was assumed to be reduced by only 2 mm due to the smaller width of the frames than in construction 1. This naturally resulted in reduced lumber savings. This gives a saving for all but the widest leaves of just under 0.2 m³/1000 pieces, which is about 2% of the material in the construction. In variant 2, the savings were mainly in particleboard, due to the much wider element width than plywood. For all leaf widths the savings were over 0.15 m³/1000 pieces. The saving on plywood was negligible, due to the small width of the elements in the leaf.

In lightweight construction 3, it was only possible to reduce the machining allowance in variant 1 by 2 mm (from 8 mm to 6 mm). The filling here was made of paper honeycomb. Reductions were made in the width of frames made of pine and MDF, with the saving in MDF being much greater, due to the larger amount of this material in the construction, and amounting to 0.15 m³/1000 pieces. On the other hand, due to the small cross-sectional dimensions of the frames, the savings expressed in percentage terms of both materials were high and amounted to almost 3%.

Construction 4 was distinguished by the use of oak wood, which is more valuable than pine. Only the lower internal frame was made of pinewood. This creates the potential for savings of this wood in variant 1 with a reduction in machining allowance from 12 mm to 8 mm. This results in a significant saving of approximately 0.4 m³/1000 pieces. In percentage terms, this is also an extremely advantageous situation, as it allows us to save

over 4% of the material in each construction variation. The saving of pinewood is negligible due to its less important role in construction 4 of the door leaf. In variant 2, particleboard savings were also significant, ranging from over 0.35 to over 0.4 m$^3$/1000 pieces, depending on the nominal width of the leaf.

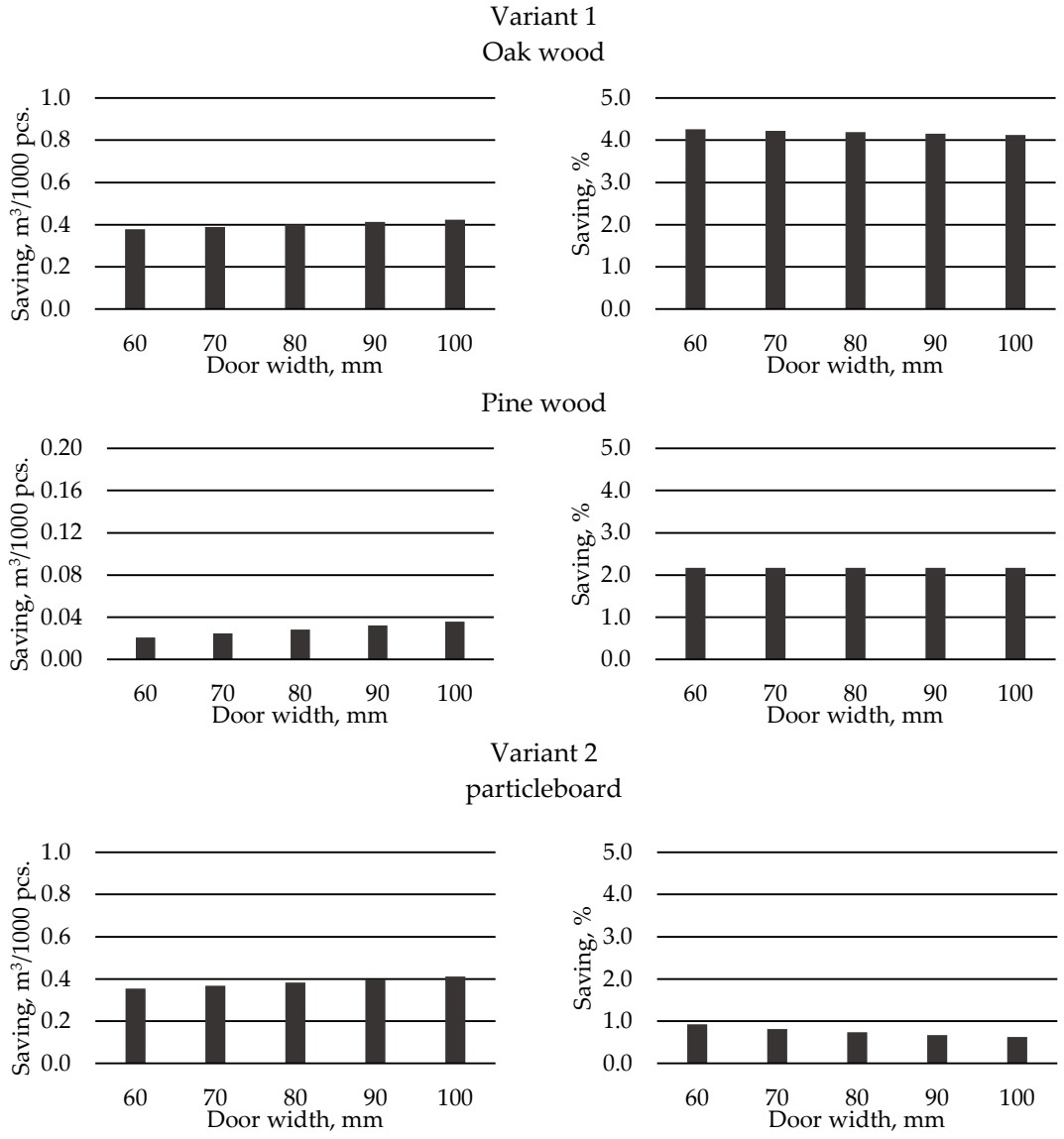

**Figure 10.** Material savings obtained by reducing the machining allowance in construction 4.

In terms of quantitative analysis of wood material savings, general observations can be made. The material savings expressed in physical units depend to some extent on the overall dimensions of the leaves. The larger they are, the more material can be saved in both variants of determining the machining allowance. In the case of savings in terms of the amount of material used in semi-finished products without reduced machining allowances, the relationship is reversed because for all leaf dimensions the value of the allowance is the same in a given construction. In each case there is also a slight saving in the use of HDF for cladding. Depending on the construction and size of the leaf, it is about 0.03 to 0.07 m$^3$/1000 pieces. Also, the overall material savings of variant 1 are slightly greater than variant 2 because they apply to the longer pieces closer to the edge of the door. Nevertheless, in this variant there are also some savings of the frame material, mainly wood, due to the reduction of their length while maintaining the transverse dimensions.

In order to verify the hypothesis concerning a significantly lower consumption of raw materials while reducing allowances, a box plot presenting the medians and deviations from them was used. It was found that at the significance level of 0.95, there are no statistically significant differences between the original material wear and the wear covered by modifications for all door widths for most structures, variants, and materials. However, significantly lower material consumption was observed after reducing the allowances for: pine wood variant 1 (construction 1), MDF (construction 3), and pine wood variant 1 (construction 4). Pine wood variant 1 (construction 1) does not deviate from the median, which means that the results are homogeneous across all door widths. It follows that reduced machining allowance in the variant 1 (reduced width of stiles) is most advantageous (Figure 11).

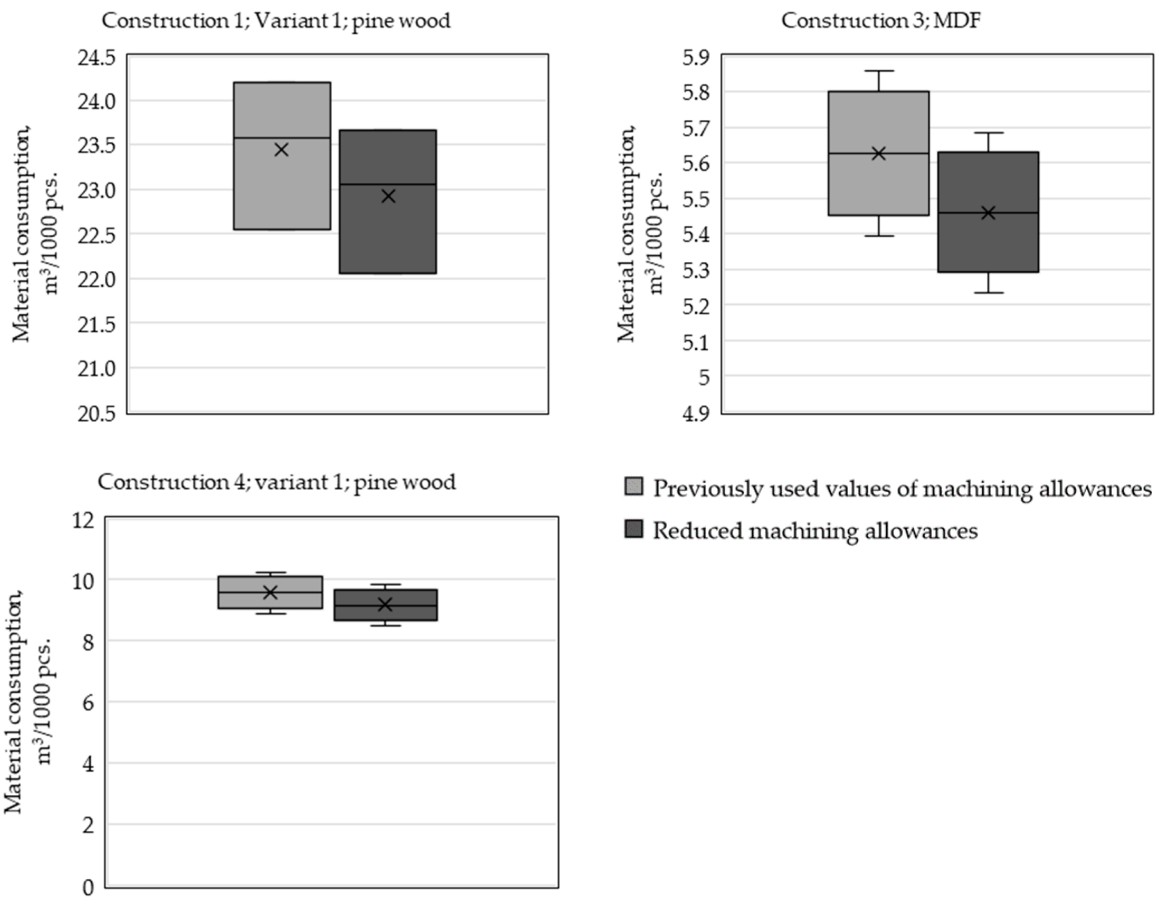

**Figure 11.** Box plots of median values of material consumption.

Considering the annual production volume of each door construction (Table 5) for the year 2020 allowed us to calculate the overall material savings. At least 43 $m^3$ of pinewood, 4 $m^3$ of oak wood, and 10 $m^3$ of MDF can be saved annually from the reduction of machining allowances in variant 1 alone. For wood, the global savings are even greater. The values calculated according to Equation (1) are 57.3 $m^3$/year and 5.3 $m^3$/year for pine and oak wood, respectively. In variant 2, particleboard savings are at least 30 $m^3$ per year.

Door leaf production in 2020 has not yet reached the design target of 800 pieces/shift within 600 shifts/year. Therefore, the target production of 480,000–500,000 pieces/year will at least double all material savings resulting from the reduction of machining allowances. The estimated total reduction in wood material consumption may eventually reach 150–200 $m^3$/year. The savings shares of different types of wood-based raw materials are shown in Figure 12.

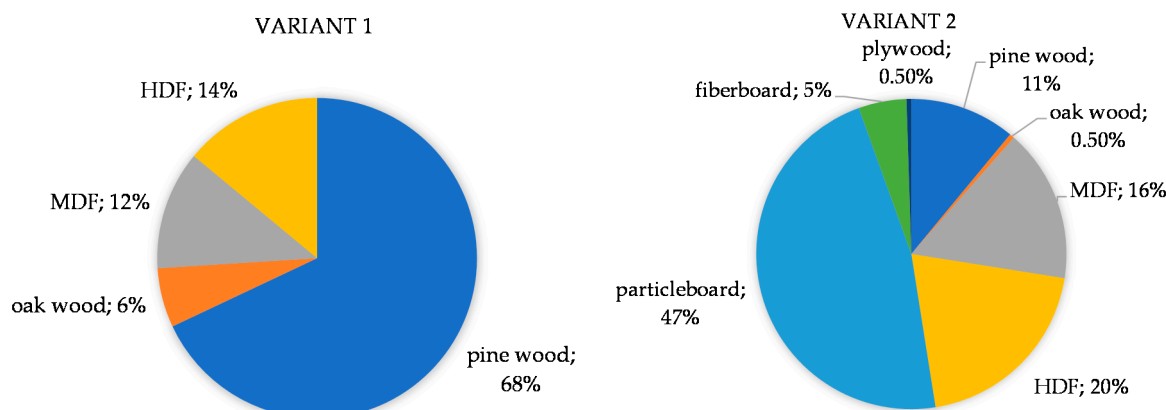

**Figure 12.** Savings by type of material.

## 5. Discussion

To analyze the competitiveness of Porta KMI Poland S.A. in the wood joinery industry it is worth using the diamond model developed by M.E. Porter (1990) which is widely recognized and valued by academia and business [29]. Within this model it is possible to understand the reason for the success of certain industries in a specific nation. The model consists of four determinants of the competitiveness of industries: factor conditions; demand conditions; related and supporting industries; and firm structure, strategy and rivalry. Moreover, there are two exogenous factors: government and chance, that also affect this interaction.

Porter explained that competition and rivalry between the companies directly affects its competitiveness. The presence of competitors leads to innovation and continuous improvement. The presence of large numbers of competitors in the cluster motivates all the firms to take notice of each others' actions and try to adopt the best strategy to face the competition. The pressure from the firms that are in proximity with each other provides inspiration to the firms to search for innovation and in turn improve their competitiveness.

TechnoPORTA's new technology line will significantly improve the company's material and cost efficiency. This is influenced by several important factors. The most intensive progress of optimization work can be observed in areas related to the elimination of unnecessary material allowances. As can be seen from the above analysis, the use of the innovative production line will allow the company to at least double all material savings resulting from the reduction of processing allowances. The estimated total reduction in the use of wood material can ultimately reach 150–200 $m^3$/year. Due to the highly precise quality of the technological process based on the proprietary TechnoPORTA Translator control system, there is a negligible number of products that do not meet processing standards and require repetition of the selected technological process (<0.5%). Improving the quality of products, and thus reducing the number of complaints, will significantly increase product margins.

In addition, the implementation of the TechnoPORTA line is in accordance with the NIS (National Intelligent Specialization) 8—Intelligent and Energy Efficient Construction standard, and in particular coincides with point I.1—materials with improved properties: structural, insulating, with increased resistance to ageing processes, vapor-permeable, with low built-in emissions, high fire resistance, low emissivity, thermo-reflective, and produced from vegetable raw materials and technologies of their production. As part of the TechnoPORTA project, an advanced technology will be developed to manufacture products meeting rigorous standards for modern construction, including fire protection, acoustic insulation, thermal insulation, burglary resistance, and to set trends for modern joinery, including high-rise buildings. Such a wide range of technical and technological possibilities will be obtainable together with the option of mass customization of orders placed by purchasers.

The TechnoPORTA line enables automation of the production process and manufacturing of leaves with dimensions and weights significantly exceeding previous technological capabilities (from 40 to even 200 kg). Until now, such a large mass and size of leaves have often hindered the processing work and caused problems with the quality and ergonomics of the work. Automation of production allows reduction of the unit cost of the product through, among other things: the ability to reduce the number of employees (line operators) and increase production efficiency (by working in 24/7 system). Automated systems allow connection of the central planning level (ERP systems, etc.) with the production level, which improves the quality of production management. Parameterized orders are automatically sent to the machines. The involvement of human capital in the production process is therefore reduced, allowing skilled workers to be redeployed to work in departments responsible for maintaining and coordinating process engineering. Automation effectively contributes to improving the quality and efficiency of the production process itself, as well as the final product—there is a reduction in costs associated with complaints or returns. Short production lines or even single-product lines are possible by placing a QR code label on each piece of the product. The retooling of the line takes place automatically.

Primary and advanced factor conditions are the obligatory inputs required by any company to compete in the market: human resources, physical resources, knowledge resources, capital resources, infrastructure resources. In the woodworking industry, the availability of skilled workers and labor costs play a key role. Another important element is the rising cost of energy, which affects the profitability of the production process and the margin of finished products. For some time now, the industry has also been struggling with limited access to raw materials such as timber, MDF, chipboard, steel sheeting, and chemical products (varnishes and adhesives), which significantly increases their prices. Reducing the consumption of materials reduces the costs associated with the purchase and storage of raw materials. Additionally, the costs related to waste management and energy consumption will decrease, which will translate into improved work safety. Ultimately, all these measures will reduce the unit technical cost of manufacturing by about 3–5%.

The presence of related and supporting industries accelerates the process of innovation and upgrades the business. Cooperation can take place along the supply chain and it gives a great potential for synergies, as partners learn from each other [30,31]. Door production requires advanced cooperation with suppliers of production and process lines, raw materials from wood, wood-based and chemical industry, IT industry. Wood and metal industry clusters are also important in the industry.

Demand conditions are fundamental elements for every company. They consist of: the structure of the demand, its size and growth patterns, and the internalization of domestic demand. If domestic customers are sophisticated and demanding as regards the product or service, they put pressure on producers to innovate faster and develop new products. This contributes to building the manufacturer's competitive advantage in the industry [32,33].

The launch of the TechnoPORTA line will enable the introduction of new collections of industrial wood-based doors to the market with the possibility of maximum consideration for the individual needs of demanding customers. This fits in with the megatrend of mass customization clearly visible in various industries. Consumers expect manufacturers to offer mass-produced products tailored to their individual, personalized needs [15,34]. This is possible due to innovations that enable the provision of goods and services that meet individual customer needs with mass production efficiency [35–37]. Customers will be able to individually specify their order within a range that significantly exceeds the existing offers of Polish door manufacturers. Each door can be personalized by, among other things, making different milling profiles, using different types and colors of veneers and edging on individual surfaces of the leaf, making oversized doors, finishing edges in postforming technology, and using any types of fittings. The application of these solutions makes it possible to introduce products with unique features and various applications to the market. It is worth noting that an important element influencing buyers' preferences is fashion in interior design. The target group of the project results' recipients are mainly

investors in the construction industry who implement multifamily residential projects, public buildings (including cultural facilities, offices, hospitals), commercial buildings, and single-family houses.

Government policy also affects the competitiveness of business entities and helps in building an efficient industry. Government policy can help to create framework conditions and rules for competition, to limit or eliminate barriers to growth, as well as to promote entrepreneurial spirit [29]. In this area, mention should be made of the need for door manufacturers to fit in with the requirements to decarbonize the economy while reducing energy demand. Manufacturers have to respect the ever-increasing requirements for ecology, segregation, and waste disposal. An example is the use of water-based rather than chemical varnishes in production.

All these factors may be beyond the control of firms. Events can affect the industry in positive ways (by creating forces that reshape the structure of an industry, allowing the firms to improve their competitive positions) as well as in negative ways (natural disasters, embargo, war) [29]. In the case of door manufacturing, the good business climate in the construction industry in recent years has played a significant role; the COVID-19 pandemic had no adverse impact on the level of demand and door sales. This was due to the increase in investment in the construction industry in Poland, which has become, and continues to be, an interesting alternative for those with financial surplus.

To sum up, it can be said that the obtained material savings in relation to the most important construction materials are an important factor influencing the competitiveness of Porta KMI Poland S.A., because it will allow the company to decrease costs related to direct material consumption. In addition, this will reduce dependence on suppliers of these materials in some way. In the technological aspect, material savings associated with the reduction of machining allowances will reduce the weight and size of semi-finished products and goods, which will contribute to the relief of employees, production and transport equipment, and storage facilities. In environmental and social terms, it will contribute to the protection of the valuable natural resources that are forests. Thus, small changes in construction, of seemingly marginal importance, broadly encompass the determinants that determine the competitiveness of industry. It is therefore worth pursuing this type of change, taking advantage of all the opportunities for innovative technology available in the industry. Only the company's activity in the field of technological and product innovation can generate opportunities to increase competitiveness in the market.

## 6. Conclusions

The new technology for the production of industrial doors, with the introduction of new product collections for sale, is an innovation on a nationwide scale. This process and product innovation, especially using customization in combination with a high, repeatable quality of products, will allow Porta KMI Poland S.A. to strengthen its competitive advantage among manufacturers of industrial doors in the domestic market. The new products will also be sold on foreign markets, thanks to which the position of Polish door manufacturers in the international arena will also be strengthened.

The implementation of the TechnoPORTA line gives the possibility to influence the volume of wood consumption in the production of industrial leaves up to 200 m$^3$/year. This is particularly the case when the reduced machining allowance refers to the width of the stiles. This translates into economic efficiency for the company, as it directly reduces material consumption. It also reduces operating costs associated with a reduction in labor intensity, loads on machinery and equipment, and inventory levels. In addition, material savings have a positive impact on the socio-economic interactions of the company by reducing dependence on suppliers of key raw materials, as well as reducing negative environmental impacts by reducing wood consumption.

**Author Contributions:** Conceptualization, Z.K. and T.R.; methodology, Z.K. and T.R.; software, T.R., Ł.S., P.S. and B.K.; validation, T.R., Z.K., J.B. and M.P.; formal analysis, T.R., Z.K., J.B. and M.P.; investigation, T.R.; resources, Z.K., Ł.S., P.S. and B.K.; data curation, Z.K., Ł.S., P.S. and B.K.; writing—original draft preparation, T.R., Z.K., J.B. and M.P.; writing—review and editing, T.R., Z.K., J.B. and M.P.; visualization, Ł.S., M.P. and T.R.; supervision, T.R.; project administration, Z.K.; funding acquisition, Z.K. All authors have read and agreed to the published version of the manuscript.

**Funding:** The results come from the project entitled "TechnoPORTA. Smart customized production line for the automated manufacture of technical doors" POIR.01.01.02-00-0095/16, co-financed within the Operational Program Smart Development, Action 1.1. R&D Projects of Enterprises', Sub-measure 1.1.2 'R&D Work related to the Production of Pilot/Demonstration Systems.

**Institutional Review Board Statement:** Not applicable.

**Informed Consent Statement:** Not applicable.

**Data Availability Statement:** Not applicable.

**Conflicts of Interest:** The authors declare no conflict of interest.

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
