# Peer review of "Innovative Line for Door Production TechnoPORTA—Technological and Economic Aspects of Application of Wood-Based Materials"

_applsci, doi:10.3390/app11104502_

Round 1

Reviewer 1 Report

Although the overall impression on this paper is very good, there are a few things that need to be addressed prior to publication.

Nowadays, companies try to save costs in various ways. Sometimes it is by increasing labor productivity, other times by saving materials, or by reducing costs while reducing too much material addition in the manufacture of doors, as in this case.

The article is written in great detail. There are too many keywords, there could be fewer. The article has many pages and it seems to me to such a range of few cited sources, despite the fact that the authors at the end of the article had a wide discussion.

The results show that there have been significant savings in materials, especially pine, oak and MDF. I lack more statistics in the results, which would be more representative and more scientific.

Reviewer 2 Report

There is a need to correct the minor deficiencies such as the identical designation of two subchapters (2.) and the like, but I especially suggest to justify why the authors refer just to the technical standards PN, CN and DIN (lines 148-149)? Why not another or more?

Furthermore, tables 1, 2, 3 and 4 (abc) are not well-arranged, seven (7) consecutive pages of tables are unbearable, such tables do not provide to the reader a direct, flexible and unambiguous overview of the materials used, the reader is sinking.

On the other hand, Door Constructions are described clearly and distinctly and their individual variants are correctly evaluated. The discussion is extensive and it is sufficient. There could be more references.

Reviewer 3 Report

This manuscript explained the implementation of the TechnoPORTA line makes it possible to reduce the negative impact on the environment by reducing the consumption of wood, aiming to study the material savings in this work. The introduction part was well written, the research gap was clearly explained (optional: the keywords could be more condensed), but some comments should be addressed:

1 As can be seen in the results, the discrepancy among the savings regards to various standards is hard to be observed, then why these standards were mentioned and compared in this work? If only one widely accepted standard was applied then the information obtained from the results may be much clearer.

2 As for the economic aspect, the author mentioned this is an innovation on the nationwide scale, then how much costs can be saved based on the novel manufacturing line?

3 The innovation of this technology should be well explained (the difference between this technology and the traditional technologies should be clarified), since this is the one of the main motivations of this work.

Reviewer 4 Report

The article is well written, and authors show deep knowledge of economics and fair engineering processes, however:

  1. The title of the article is not connected with the aim of the article. Maybe an appropriate title should be “Innovative technical line for Door Production Line TechnoPORTA - …”
  2. There are not enough details regarding exactly what were the novelty techniques and how to reproduce them by other people to verify the achieved and reported results. Values are showing newer machining allowances, but how were the new values were achieved?
  3. This study is based on estimates and not from obtained data. There are insufficient statistical data to sustain the theory that the new implementations are representative of the estimate.
  4. Tables and plots are excessive and with non-relevant data. Showing the new technique details would be more important (see #1) than this bunch of information. Furthermore, reporting the saved materials by the type of material would be a good way to present the results. Tables like that would be important to discuss discrepancies, not similarities.

As it is, there is no scientific contribution – saving material is the focus of every single manufacturing company. Even if details are given (such as engineering schemes of the new machining aspects) this report is not scientific but an advertisement – this is clear in the first paragraph of the conclusions. Reading as an Industrial Engineer it is possible to understand how the reported information is important, but as a Wood Scientist, no significant contribution is found on this paper. A major rewriting is necessary.

Round 2

Reviewer 4 Report

Dear authors,

I'm satisfied with your replies to my review. I'm glad you could provide all the statistical data. This addition totally changed the weight and value of your publication.

The explanation regarding the automation and sensors used to control the allowances is very good, and the new pictures will be appreciated by your readers. This addition brought scientific contribution and repeatability to your study.

Regarding the tables and plots, I understand you are following a Polish standard. I do not think you need to change it now, but I suggest you to, if you have another publication in the future where you also have a mandatory and huge amount of data, add these tables to support files/attachments. I appreciated the fact you added plots and figures to condensate and contrast the results at the end of those sections. 

The new plot reporting the saved materials by their type is outstanding. There is a huge contribution to the Environment. I appreciate your efforts in reporting these results.